# Comprehensive germline genomic profiles of children, adolescents and young adults with solid tumors

Sara Akhavanfard[1,2], Roshan Padmanabhan [1], Lamis Yehia [1], Feixiong Cheng[1,2,3] & Charis Eng [1,2,3,4,5 ✉]

Compared to adult carcinomas, there is a paucity of targeted treatments for solid tumors in children, adolescents, and young adults (C-AYA). The impact of germline genomic signatures has implications for heritability, but its impact on targeted therapies has not been fully appreciated. Performing variant-prioritization analysis on germline DNA of 1,507 C-AYA patients with solid tumors, we show 12% of these patients carrying germline pathogenic and/ or likely pathogenic variants (P/LP) in known cancer-predisposing genes (KCPG). An additional 61% have germline pathogenic variants in non-KCPG genes, including *PRKN*, *SMARCAL1*, *SMAD7*, which we refer to as candidate genes. Despite germline variants in a broad gene spectrum, pathway analysis leads to top networks centering around p53. Our drug-target analysis shows 1/3 of patients with germline P/LP variants have at least one druggable alteration, while more than half of them are from our candidate gene group, which would otherwise go unidentified in routine clinical care.

[1] Genomic Medicine Institute, Lerner Research Institute, Cleveland Clinic, Cleveland, OH 44195, USA. [2] Cleveland Clinic Lerner College of Medicine, Case Western Reserve University, Cleveland, OH 44195, USA. [3] Cancer Prevention, Control & Population Research Program, Case Western Reserve University School of Medicine, Cleveland, OH 44106, USA. [4] Germline High-Risk Cancer Focus Group, Case Comprehensive Cancer Center, Case Western Reserve University School of Medicine, Cleveland, OH 44106, USA. [5] Department of Genetics and Genome Sciences, Case Western Reserve University School of Medicine, Cleveland, OH 44106, USA. ✉email: engc@ccf.org

Solid tumors account for half of the malignancies in children, adolescents, and young adults (C-AYA), with a lower burden of somatic variants, and are assumed to have higher frequencies of germline alterations, compared to adults with solid tumors[1,2]. Although there has been substantial advancement in understanding somatic variants in cancers, our knowledge regarding the spectrum, frequency, and implications of germline variants in C-AYA with solid tumors is limited. Recent pan-cancer studies showed that 7–8% of patients with these malignancies diagnosed <20 years of age have pathogenic or likely pathogenic (P/LP) germline variants in known cancer-predisposing genes, with adrenocortical carcinoma (50%) and high-grade glioma (25%) having the highest percentage of variants among all solid tumors. However, these percentages could be biased for the following reasons: a high proportion of patients with those specific tumor types and the absence of a few other subtypes of pediatric cancers in those studies. In contrast, 1.1% of the control group studied by Zhang et al.[1–4], comprised of healthy adult individuals from the 1000 Genomes Project, carry such variants.

The cancer-predisposing genes mainly encode tumor suppressors, which are related to DNA damage to ensure DNA repair processes or oncogenes, which promote growth. In general, pathogenic loss-of-function variants in oncogenes can disrupt normal cellular processes, and predispose to cancer development[5]; moreover, multiple genes can have both types of variants with different functional and phenotypic effects. It is well established that there is an overlap between germline cancer-predisposing genes and somatic tumor-driver genes: there are many examples showing identical genes having roles in somatic oncogenesis and susceptibility to cancer, respectively[6].

In pediatric and AYA clinics, family history is essentially the primary means used to recognize patients with possible heritable cancer[7]. This is despite prior studies showing that a family history of cancer could only be obtained in about 40% of patients with P/LP mutations due to multiple limitations[2]. It is now confirmed, as well, that there is a remarkably elevated risk of secondary primary neoplasms in C-AYA cancer survivors who carry a germline P/LP mutation in cancer-predisposing genes compared to those who do not[8]. In addition to implications for heritability and second primary neoplasms, germline (in every single cell of the body) variants can also provide novel therapeutic targets. The clonal nature of germline variants compared to the heterogeneous somatic pattern of tumors make them potentially a suitable biomarker and therapeutic target, both of which are lacking for C-AYA malignancies, compared to adult malignancies[9]. Here, we address this gap by investigating germline genomic signatures of 1507 patients with solid tumors diagnosed under 29 years of age.

## Results

**Germline alterations in Cleveland Clinic patient series**. We evaluated 50 prospectively enrolled C-AYA patients at the Cleveland Clinic (CCF), with a broad range of solid tumors diagnosed under 29 years of age. The series had a median age of $12 \pm 7.1$ years (range 0.5–29) and consisted of 31 children (52% females, median age of $8 \pm 4.2$ years), 12 adolescents (66.7% males, median age of $18 \pm 1.2$ years), and 7 young adults (all males, median age of $22 \pm 3.6$ years). Collectively, these patients had 14 different tumor types, with bone and soft tissue sarcomas being the predominant cancer types (Supplementary Table 1; Supplementary Data 1).

First, we analyzed 204 known cancer-predisposing genes (KCPG), curated using previously established cancer-predisposing genes in addition to the newly proposed genes from recent publications (Supplementary Data 2; Supplementary Fig. 1). We found three pathogenic germline variants (Methods), one nonsense mutation in TP53, and two frameshift indels in BRCA2 and GJB2 genes in two patients with osteosarcoma, which were further confirmed by Sanger sequencing (Table 1; Fig. 1a, b; Supplementary Data 3; Supplementary Fig. 2). The average mean depth was 258× (range 45×–444×) for the CCF P/LP KCPG variants. Assessing germline copy number variations (CNVs), using exome coverage data, we found five genes with germline duplications, including DDX10 and SUZ12 (Supplementary Fig. 3; Supplementary Data 4). There were no known CNVs in the identified regions in the database of genomic variants (DGV). In a rare circumstance, a 27-year-old male with multiple primary sarcomas was found to have two pathogenic KCPG variants, one in BRCA2 (paternally inherited) and the other in TP53 (maternally inherited), the latter confirming a Li–Fraumeni syndrome diagnosis (Table 1). Both parents are in their 50s with no history of cancer. Our second representative case was a female patient with osteosarcoma, diagnosed at 10, who carried a pathogenic variant in GJB2 in addition to a germline duplication of DDX10, the latter, a known marker somatically associated with poor prognosis for osteosarcoma (Table 1; Fig. 1c; Supplementary Fig. 3a). Overall, 2 out of 50 C-AYA CCF patients with solid tumors carried a germline pathogenic KCPG variant, and 3 other C-AYA CCF patients harbored a germline CNV.

Evaluation of pedigrees to obtain family histories revealed the existence of a positive family history of cancer in about 40% of the remaining 42 (84%) patients. This suggests the existence of yet-to-be-identified predisposing genes in KCPG mutation-negative patients. Hence, we extended our analyses to explore other P/LP variants from non-KCPG, which we will refer to as candidate genes. Our variant classification based on the ACMG guidelines identified 59 predicted pathogenic and 37 predicted likely pathogenic variants in 89 candidate genes (Fig. 1a, b; Supplementary Data 3). The average mean depth was 163× (range 20×−454×) for all the CCF P/LP candidate variants. Overall, 34 out of the remaining 45 patients

---

**Table 1 CCF patients with germline alterations in known cancer-predisposing genes.**

| Case# | Sex, age | C-AYA tumors | Syndrome | Gene | Germline alterations |
|---|---|---|---|---|---|
| CCF12237 | M, 27 years | Chondroblastic osteosarcoma of the left maxilla, 22 years | Li–Fraumeni syndrome | TP53 | NM_000546.5; c.916C>T (p.Arg306*) |
| | | Odontoameloblastoma of the left mandible, 24 years | | BRCA2 | NM_000059.3; c.4284dupT (p.Gln1429Serfs*9) |
| | | Leiomyosarcoma of the right scrotum, 25 years | | | |
| CCF11829 | F, 17 years | Osteosarcoma, 10 years | None | GJB2 | NM_004004; c.35delG (p.G12fs*2) |
| | | | | DDX10 | NM_004398; duplication (Chr11:108535752-108811657) |

C-AYA children, adolescents, and young adults.

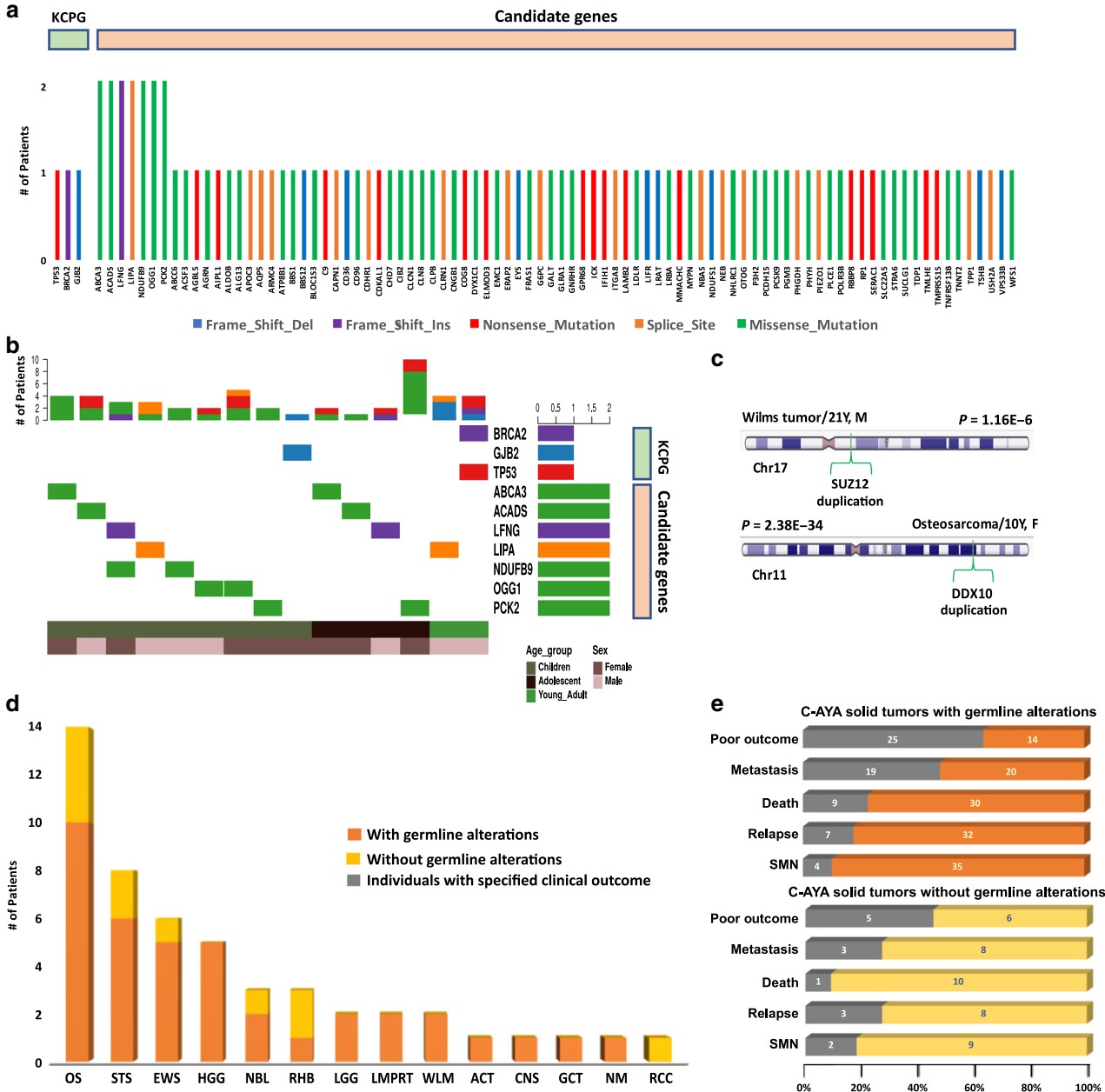

**Fig. 1 Germline alterations and clinical outcomes in the Cleveland Clinic series. a** Genes with germline pathogenic/likely pathogenic (P/LP) variants in known cancer-predisposing genes (KCPG) and candidate genes and their type of alterations in children, adolescents, and young adult (C-AYA) patients with solid tumors. **b** Oncoplots of top mutated genes with P/LP variants in KCPG and candidate genes based on the age group. Each column represents one patient and its affected genes. **c** Two examples of copy number variations (CNVs) found in C-AYA patients with solid tumors. **d** The number of patients with germline alterations, both single-nucleotide variations (SNVs) and CNVs, in each tumor type. **e** Clinical outcome comparison between two groups of C-AYA patients with solid tumors, with and without germline alterations. Gray color represents the number of patients with the specified clinical outcome in each group. Two-sided Fisher's exact test was implemented, $P = 0.31$, OR = 2.14, 95% CI = 0.6–8.0. ACT adrenocortical carcinoma, CNS central nervous system, EWS Ewing sarcoma, GCT germ cell tumor, HGG high-grade glioma, LGG low-grade glioma, LMPRT low malignancy potential renal tumor, NBL neuroblastoma, NM non-malignant tumor, OS osteosarcoma, RCC renal cell carcinoma, RHB rhabdomyosarcoma, STS soft tissue sarcoma, WLM Wilms tumor, Del deletion, Ins insertion.

(75.5%) had at least one variant in a candidate gene (Fig. 1d). C-AYA patients with germline alterations trended towards a poor outcome which we operationally defined as any relapse, metastasis, second primary malignant neoplasm (SMN), or death, compared to the group without germline alterations (Fig. 1e) ($P = 0.31$, odds ratio (OR) = 2.14, 95% confidence interval (CI) = 0.6–8.0).

To validate our findings with a larger independent series, we then analyzed germline exome data from 1457 C-AYA patients with solid tumors from the St. Jude (StJ) dataset. Because we

roughly saw similar mutation patterns in the StJ dataset as our CCF series, we combined the CCF and StJ series for further analyses. The combined dataset included 1507 patients with a median age of $6.41 \pm 5.8$ years consisting of 1182 children (50.7% females, median of $5.2 \pm 4.5$ years), 164 adolescents (59.1% males, median age of $16.8 \pm 1.3$ years), 20 young adults (75% males, median age of $21 \pm 2.4$ years), and 141 unknown age group who were diagnosed with solid tumors under 29 years of age. The most common tumor types included CNS tumors in 323 patients

(21.4%), followed by Wilms tumors in 207 patients (13.7%), neuroblastomatas in 190 patients (12.6%), and rhabdomyosarcomas in 134 patients (8.9%). Compared to the predicted frequencies of solid tumors in the Surveillance, Epidemiology, and End Results (SEER) program (http://seer.cancer.gov/iccc), our study had overrepresented numbers of cases with Wilms tumor, retinoblastoma, and osteosarcoma (1.8–2.1 times more), and underrepresented numbers of cases with germ cell tumors and adrenocortical carcinomas (two times less) (Table 2).

**Germline variants in known cancer predisposition genes.** In analyzing 204 KCPG (Methods; Supplementary Data 2), we found 158 pathogenic and 40 likely pathogenic variants in 182 patients (12%). The average mean depth was 120× (range 21×−410×) for all the P/LP KCPG variants. The average variant allele fraction in P/LP KCPG variants was 47% (±9.2). The pathogenic variants included 53 frameshift indels (37 deletions, 16 insertions), 53 nonsense, 25 splice-site, 23 missense, 3 start-loss mutations, and 1 in-frame deletion (Supplementary Data 5–6; Supplementary Fig. 4). The majority of the likely pathogenic variants (37, 92.5%) were missense mutations (Supplementary Data 5). *RB1* (32 patients, 53% with nonsense mutations), *NF1* (22 patients, 41% with nonsense mutations), *CHEK2* (19 patients, 58% with frameshift deletions), and *TP53* (10 patients, 50% with missense mutations) were the genes with the most frequent P/LP mutations among the 54 mutated genes in our dataset (Fig. 2a, b). All of these 198 P/LP variants belong to KCPG genes with autosomal-dominant (AD), autosomal-recessive/autosomal-dominant (AR/AD), or X-linked-dominant (XLD) pattern of inheritance. We excluded all the autosomal-recessive KCPG variants since we only identified heterozygous alterations (Supplementary Data 7). From 182 patients with P/LP KCPG mutations, 168 (92.3%) individuals had only one P/LP KCPG variant each. Twelve patients carried two P/LP KCPG variants, with 6 of those diagnosed with retinoblastoma. We had a 2-year-old male patient with adrenocortical carcinoma who had three P/LP KCPG variants in *NTRK1*, *EP300*, and *HMBS* genes. Our second case with three P/LP KCPG variants was a female diagnosed with CNS tumor at 6 years of age and carrying variants in *NF1*, *LZTR1*, and *RECQL* genes; she is a cancer survivor with no SMN after 20 years of clinical follow-up (Supplementary Data 8)[8].

**Germline variants in candidate genes.** Beyond KCPG variants, we identified 1825 pathogenic and 896 likely pathogenic variants in 1173 candidate genes (Fig. 2a, b; Supplementary Data 9; Supplementary Fig. 5). This includes an additional 925 (61%) patients with predicted pathogenic variants and 193 (13%) patients with likely pathogenic variants. The average mean depth was 99× (range 20×−1040×) for all the P/LP candidate variants. One thousand one hundred one (40%) of predicted pathogenic and 380 (42%) of predicted likely pathogenic variants had loss-inferred-activity by IVA analysis (Supplementary Data 9). The average variant allele fraction in the candidate P/LP variants was 47% (±9.4). Candidate P/LP variants included 696 nonsense, 418 splice-site, 377 frameshift indels (197 deletions, 180 insertions), 1179 missense, and 51 in-frame indels (44 deletions, 7 insertions) (Supplementary Data 9). Among 1173 mutated candidate genes in our dataset, *PRKN* (23 patients), *PAH* and *TYR* (each found in 17 patients), and *EYS* and *TMPRSS3* (each found in 16 patients) had the highest number of P/LP variants (Fig. 2a, b; Supplementary Data 10–11). As a control, the same variant analysis was performed on data from 53,105 individuals from the Exome Aggregation Consortium dataset (ExAC), excluding individuals belonging to The Cancer Genome Atlas (TCGA), known as non-TCGA ExAC dataset. Overall, 28% of the candidate genes with four and more P/LP variants had statistically significant P/LP variant allele frequencies (OR = 4.3–247 and infinity, Bonferroni-corrected $P$ values = 0.049 to $2.44 \times 10^{-17}$) in our C-AYA dataset compared to that in the non-TCGA ExAC dataset (Supplementary Data 12–13). Interestingly, this percentage was equal to the one calculated for the KCPG group with four and more P/LP variants versus non-TCGA ExAC dataset (28%, odds ratio = 7.4–40.7, Bonferroni-corrected $P$ values = 0.039 to $3.16 \times 10^{-31}$).

**Table 2 Demographics and clinical characteristics of patients.**

| Source | CCF/PCGP/SJLIFE | | | | |
|---|---|---|---|---|---|
| Age group | Children | Adolescent | Young adult | Unknown | Total |
| Female/Male | 600/582 | 67/97 | 5/15 | 14/13[a] | 686/707 |
| Mean age of onset | 6.1 ± 4.5 | 16.9 ± 1.3 | 22 ± 2.4 | NA | 7.6 ± 5.8 |
| All solid tumors (1507) | 1182 | 164 | 20 | 141 | 1507 |
| Central nervous system (323) | 266 | 23 | 2 | 32 | 323 |
| Wilms tumor (207) | 189 | 2 | 1 | 15 | 207 |
| Neuroblastoma (190) | 158 | 2 | 1 | 29 | 190 |
| Rhabdomyosarcoma (134) | 114 | 14 | 1 | 5 | 134 |
| Osteosarcoma (129) | 78 | 42 | 5 | 4 | 129 |
| Retinoblastoma (98) | 84 | | | 14 | 98 |
| Ewing's sarcoma (95) | 58 | 27 | 5 | 5 | 95 |
| Soft tissue sarcoma (93) | 69 | 17 | 3 | 4 | 93 |
| High-grade glioma (80) | 63 | 8 | | 9 | 80 |
| Germ cell tumor (74) | 57 | 11 | 1 | 5 | 74 |
| Low-grade glioma (24) | 8 | 1 | | 15 | 24 |
| Adrenocortical carcinoma (22) | 20 | 1 | | 1 | 22 |
| Carcinoma (14) | 7 | 7 | | | 14 |
| Giant cell tumor (3) | 0 | 3 | | | 3 |
| Renal cell carcinoma (3) | 2 | | 1 | | 3 |
| Low malignant potential renal tumors (2) | 2 | | | | 2 |
| Basal cell carcinoma (1) | 1 | | | | 1 |
| Paraganglioma (1) | | 1 | | | 1 |
| Other solid tumor (10) | 3 | 4 | | 3 | 10 |
| Non-malignant tumor (4) | 3 | 1 | | | 4 |

*CCF* Cleveland Clinic Foundation, *PCGP* Pediatric Cancer Genome Project, *SJLIFE*, St. Jude Life Cohort. [a]Gender data were not available for all the patients in this group.

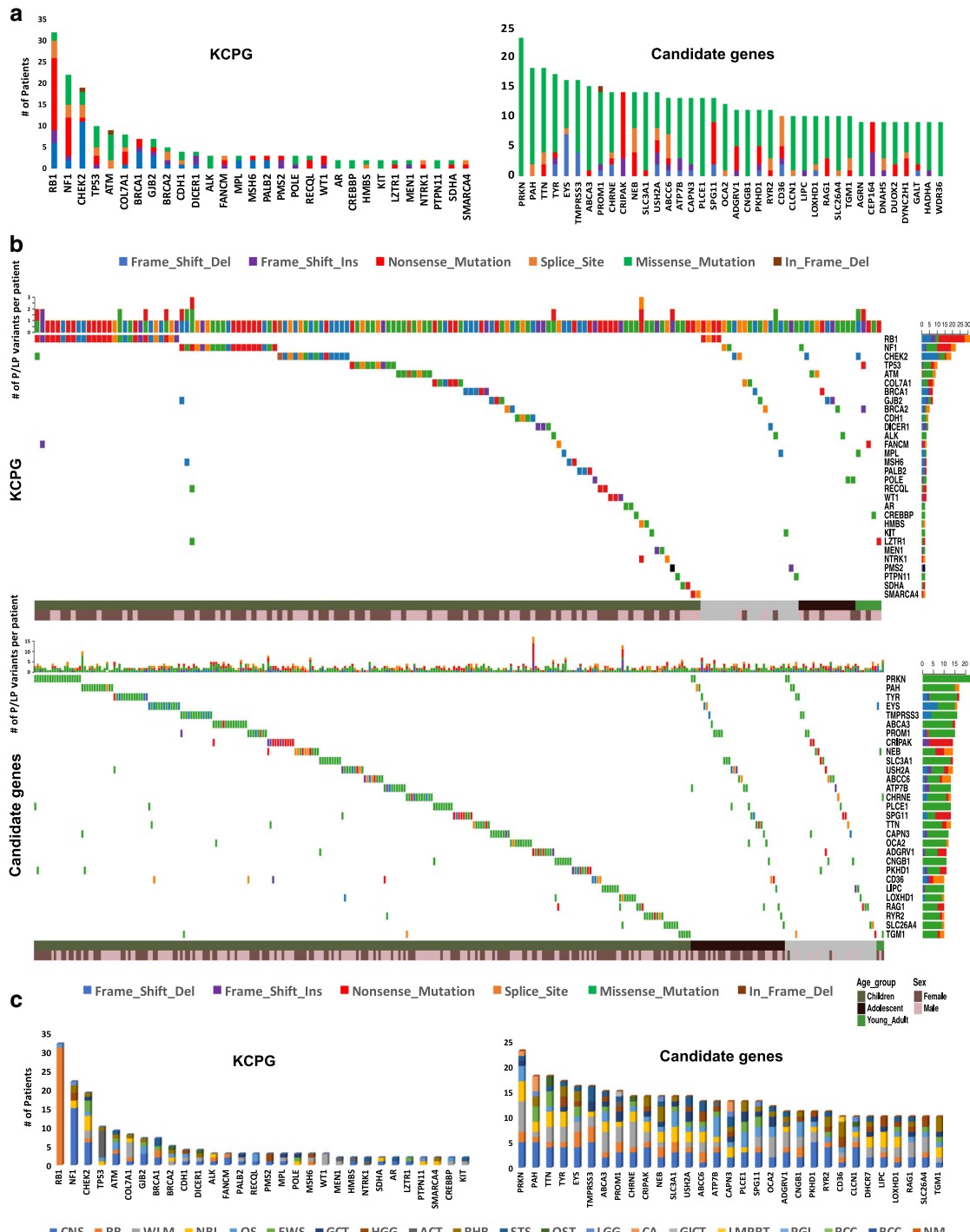

**Fig. 2 Distribution of germline pathogenic/likely pathogenic (P/LP) mutations in children, adolescents, and young adults (C-AYA) with solid tumors.**
**a** Top mutated genes with germline P/LP variants in KCPG (left panel) and candidate genes (right panel) and their type of alterations. **b** Oncoplots of top mutated genes with P/LP variants in known cancer-predisposing genes (KCPG) (top panel) and candidate genes (lower panel) based on the age group. Each column represents one patient and its affected genes. **c** Most frequently mutated genes with P/LP variants in KCPG (left panel) and candidate genes (right panel) based on their affected tumor types. ACT adrenocortical carcinoma, BCC basal cell carcinoma, CA carcinoma, CNS central nervous system, EWS Ewing sarcoma, GCT germ cell tumor, GICT giant cell tumor, HGG high-grade glioma, LGG low-grade glioma, LMPRT low malignancy potential renal tumor, NBL, neuroblastoma, NM non-malignant tumor, OS osteosarcoma, OST other solid tumors, PGL paraganglioma, RB retinoblastoma, RCC renal cell carcinoma, RHB rhabdomyosarcoma, STS soft tissue sarcoma, WLM Wilms tumor, Del deletion, Ins insertion.

**Germline genomic signatures across solid tumor types**. To find a germline genomic signature for each tumor type, we classified our germline data by individual tumor type, for 12 specific types of solid tumors (samples ranging from 22 to 323 cases per tumor type). Individuals with adrenocortical carcinoma and high-grade

glioma tumors had the highest number of germline P/LP variants (combined KCPG and candidate genes) per sample, 3.6 and 2.8, respectively, compared to the overall 1.9 P/LP variants per sample. Half of the individuals with ACT and 45% of retinoblastoma cases carried at least one KCPG P/LP variant per patient. While

each type of C-AYA solid tumor had its own well-known associated germline KCPGs, we reported unexpected KCPGs for many tumor types (Figs. 2c and 3a; Supplementary Data 14–15). For example, while TP53, PMS2, and RET are already reported as genes with germline alterations in individuals with Ewing sarcoma, we identified germline P/LP variants in ATM (1 patient), BRCA1 (1), CHEK2 (3), GJB2 (1), LZTR1 (1), and POLE (1) genes in cases with Ewing sarcoma (Supplementary Fig. 6). Other examples include germline P/LP variants in MEN1 (1), BRCA2 (1), PALB2 (1), KIT (1), MPL (1), CDC73 (1), and COL7A1 (4) in patients with Wilms tumor. Although RB1 was mutated in about

one-third of our retinoblastoma cases, RB1 mutation-negative retinoblastoma patients had germline P/LP variants in other known cancer predisposition genes like BRCA1 (2), EGFR (1), and MSH6 (1) (Fig. 3a, b).

Beyond P/LP KCPG variants, we demonstrated an interesting signature of germline P/LP variants in our candidate gene group (Fig. 3c, d). For example, we found germline P/LP variants in TMPRSS3, a member of the serine protease family, in patients with CNS tumors (5), retinoblastoma (3), and soft tissue sarcoma (3) (Fig. 3c, d). Another example is the detection of germline P/LP variants in MCPH1, which encodes a DNA damage

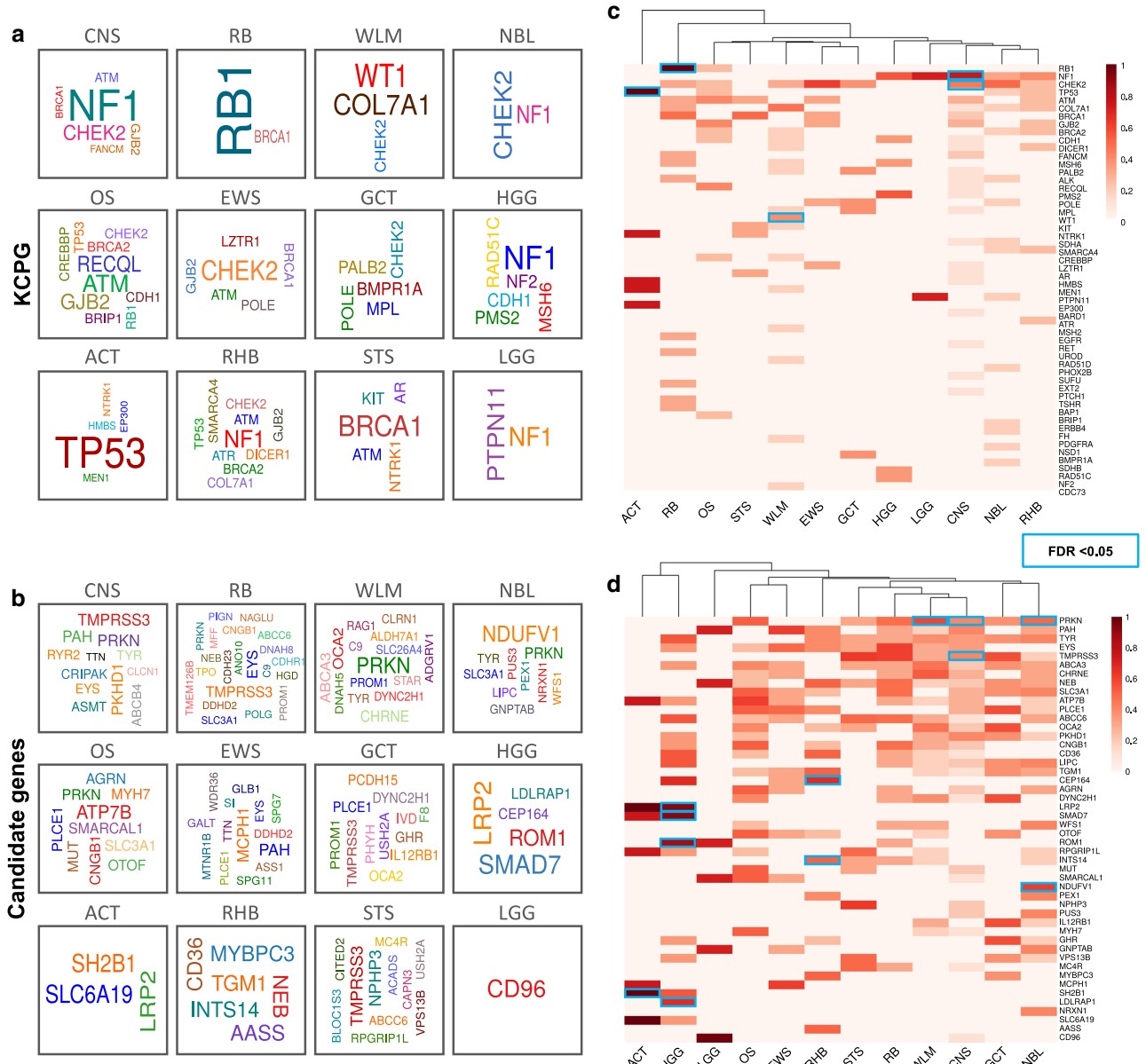

**Fig. 3 Germline genomic signatures of children, adolescents, and young adults (C-AYA) with 12 types of solid tumors. a**, **b** Germline gene cloud signatures of the C-AYA patients with solid tumors based on their altered known cancer-predisposing genes (KCPG) (**a**) or candidate genes (**b**). The size of the genes is proportional to their pathogenic/likely pathogenic (P/LP) variant frequency in that tumor type, colors do not specify any meaning. **c**, **d** Heat maps of top altered KCPGs (**c**) or candidate genes (**d**). Two-sided Fisher´s Exact test implemented in R statistical software. *P* values were adjusted for multiple testing with Bonferroni correction considering 593 tests. FDR threshold of 0.05 considered a significant event. Scale refers to log10 (frequency of P/LP variants in specified genes in each tumor type). Blue rectangles specify significant corrected *P* values in comparison to non-TCGA ExAC database. ACT adrenocortical carcinoma, CNS central nervous system, EWS Ewing sarcoma, GCT germ cell tumor, HGG high-grade glioma, LGG low-grade glioma, NBL neuroblastoma, OS osteosarcoma, RB retinoblastoma, RHB rhabdomyosarcoma, STS soft tissue sarcoma, WLM Wilms tumor.

response protein in three of our Ewing sarcoma patients. Germline P/LP variants in *SMARCAL1*, another gene related to the DNA damage pathway, was detected in three patients with osteosarcoma. Germline P/LP variants in *SMAD7*, a gene associated with colorectal and breast cancers, were found in six of our high-grade glioma patients. Germline loss of function of the *PRKN* gene was frequently found in our patients with Wilms (6), CNS (5), neuroblastoma (4), and osteosarcoma (3) tumors (Fig. 3c, d). Our top proposed candidate genes selected based on their known connection with cancers, and their predicted mechanisms of function can be found in Table 3.

Finally, in order to investigate if C-AYA patients with solid tumors have any germline alteration enrichment in genes related to other common non-cancerous congenital syndromes, we cross-matched our data with 185 congenital heart defect (CHD)-related genes and found 67 (4.4%) of our patients carried at least one P/LP variant in one autosomal-dominant CHD-related gene (7 KCPG and 19 candidate genes). An additional 72 (4.8%) patients carried heterozygous variants in autosomal-recessive CHD-related genes (2 KCPG and 31 candidate genes), which we did not include in our analysis due to their mode of inheritance. C-AYA patients with CNS (32 variants), neuroblastoma (21), rhabdomyosarcoma (15), and Wilms (15) tumors had the highest number of CHD-related gene variants, and patients with retinoblastoma (2) and osteosarcoma (2) tumors had the least number of those variants (Supplementary Fig. 7; Supplementary Data 16).

**Pathway analysis across all solid tumors**. Since we noticed a very broad spectrum of both KCPG and candidate genes involved in C-AYA solid tumors, we next sought to determine if they converged on any common pathways. The p53 pathway with fraction affected of 0.5 (3 out of 6 genes) was the most affected, followed by receptor tyrosine kinases and Ras (RTK–RAS) pathway with fraction affected of 0.14 (12 out of 85 genes), Hippo pathway with fraction affected of 0.13 (5 out of 38 genes) (Fig. 4a;

Supplementary Data 17). Candidate genes were remarkably involved in the affected pathways. For example, in the RTK–RAS pathway, 3 out of 12 mutated genes were from the candidate gene group, including *SHC1*, *ERBB3*, and *FLT3* (Fig. 4b). The Hippo pathway had four affected candidate genes, *CRB1*, *CRB2*, *HMCN1*, and *LATS1*. In the Wnt pathway, we had one recently recognized KCPG, *LZTR1*, and the other six affected genes were from the candidate gene group: *AXIN1*, *WNT10A*, *CHD8*, *FZD6*, *LRP5*, *RSPO1*. Although only two genes belonged to the Cell Cycle Pathway, *RB1*, one of those two, was the gene with the highest variant frequencies in our dataset with 32 cases (24 variants) (Supplementary Fig. 8).

Other than the direct effect on each pathway, we investigated the interactions between our mutated genes through Ingenuity Pathway Analysis (IPA) (Supplementary Data 18). Only genes with at least four variants in the dataset were used to generate our networks. The connection between our target genes and molecules in the IPA knowledge database formed the basis of this network construction. Our IPA-predicted top network comprised 26 of our genes (10 KCPG and 16 candidate genes) centering around p53 (right-tailed Fisher's exact test $P = 1 \times 10^{-42}$; Fig. 4c). Top diseases and functions predicted to be affected by this network were metabolic diseases, organismal injury and abnormalities, and cancer. Our analysis showed the top anticipated canonical pathways affected by our target genes were DNA double-strand break repair by homologous recombination (28.6% overlap, Benjamini–Hochberg (B–H) corrected $P = 2.08 \times 10^{-3}$), role of BRCA1 in DNA damage response (11.2% overlap, B–H corrected $P = 5.82 \times 10^{-5}$), and role of CHK proteins in cell cycle checkpoint control (8.8% overlap, B–H corrected $P = 2.48 \times 10^{-2}$). Using our network analysis, we could predict that Eukaryotic Translation Initiation Factor 4 Gamma 1 (EIF4G1, B–H corrected $P = 1.39 \times 10^{-3}$) and I kappa b kinase (IκB kinase, B–H corrected $P = 1.39 \times 10^{-3}$) could act as master upstream regulators for the altered genes and potentially control the expression of those altered genes (Fig. 4d).

**Table 3 Ten examples of candidate genes and their associations with cancer.**

| Candidate genes | OMIM | Related cancers | Non-cancerous syndrome/disease | Mechanism of function | References |
|---|---|---|---|---|---|
| *SMAD7* | 602932 | Familial colorectal cancer; pancreatic cancer | None | Via inhibiting TGF-b signaling | 58 |
| *PRKN* | 602544 | Hereditary breast and/or ovarian cancer, lung cancer | Familial Parkinson disease | Via regulating PI3K/AKT pathway via inactivation of PTEN | 22,24 |
| *TYR* | 606933 | Hereditary melanoma; basal cell carcinoma | Oculocutaneous albinism | Via dysregulation of melanin synthesis | 59 |
| *GHR* | 600946 | Hereditary breast cancer | Familial cardiovascular disease | Via GH/IGF-1 pathway | 60 |
| *SAMD9* | 610456 | Familial normophosphatemic tumoral calcinosis; inherited myelodysplastic syndromes; breast and colon cancers | Hereditary connective tissue disorder | Via impairing endosomal function | 61,62 |
| *TMPRSS3* | 605511 | Breast, ovarian, and pancreatic cancers | Hereditary sensorineural hearing loss | Via regulating ERK1/2 and PI3K/Akt pathways | 16,18 |
| *SMARCAL1* | 606622 | Clear-cell renal cell carcinoma; endometrioid cancer | Hereditary connective tissue disorder | Via defects in DNA damage repair/cell cycle checkpoints | 63,64 |
| *ABCB4* | 171060 | Lung, breast, head and neck, skin and cervix cancers; cholangiocarcinoma | Progressive familial intrahepatic cholestasis | Via genome instability and copy number gains in the MAPK signaling pathway | 65,66 |
| *MCPH1* | 607117 | Hereditary breast cancer; ovarian cancer | Hereditary connective tissue disorder | Via causing mitotic errors its involvement in the spindle checkpoint and apoptosis | 67,68 |
| *MERTK* | 604705 | Glioblastoma; rhabdomyosarcoma; breast, colon, and gastric cancers | Hereditary retinal degeneration | Via regulating ERK1/2 and PI3K/Akt pathways | 69 |

*GH* growth hormone, *IGF-1* insulin-like growth factor-1, *TGF-b* transforming growth factor beta, *AKT* protein kinase B, *PTEN* phosphatase and tensin homolog, *ERK* extracellular signal-regulated kinase, *MAPK* mitogen-activated protein kinase, *PI3K* phosphoinositide 3-kinase.

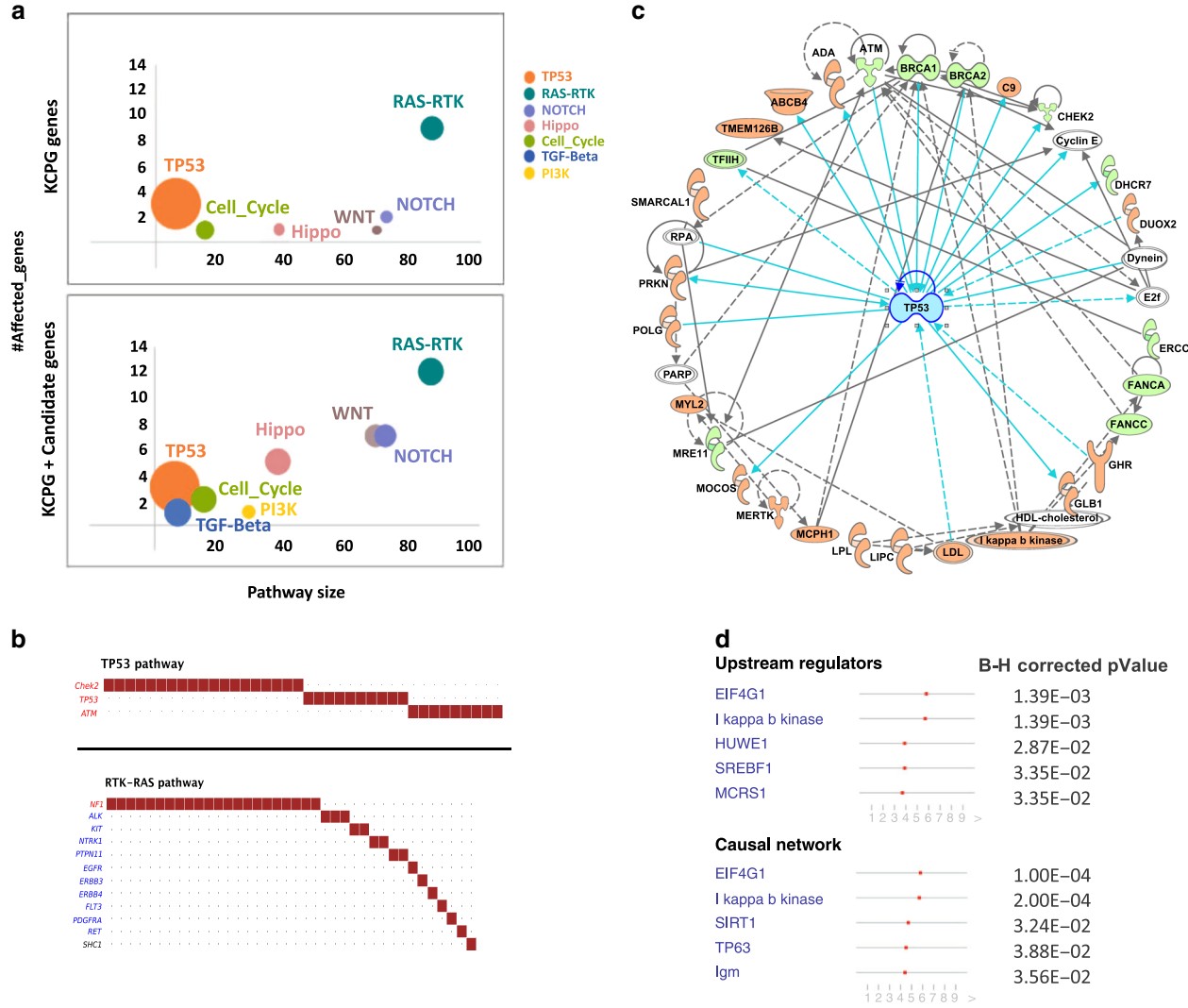

**Fig. 4 Pathway analysis of altered genes with germline pathogenic/likely pathogenic (P/LP) variants in children, adolescents, and young adults (C-AYA) with solid tumors. a** Affected pathways based on altered genes with P/LP germline variants. Top panel: only known cancer-predisposing genes (KCPG), lower panel: a combination of all KCPGs and candidate genes. Size of the circles increases as the fraction affected increases. **b** Genes mutated in TP53 (top panel) and RAS-RTK (lower panel) pathways, and the number of patients affected in our cohort. Red font: tumor suppressor genes; blue font: oncogenes. **c** Top network, predicted by Ingenuity Pathway Analysis (IPA), based on all the KCPG (green color) and candidate genes (salmon color) with at least four P/LP variants in our C-AYA patients with solid tumors (right-tailed Fisher's exact test $P = 1 \times 10^{-42}$). **d** Eukaryotic Translation Initiation Factor 4 Gamma 1 (EIF4G1, B-H corrected $P = 1.39 \times 10^{-3}$) and I kappa b kinase (IκB kinase, B-H corrected $P = 1.39 \times 10^{-3}$) predicted to be the top upstream regulators/causal network based on our IPA analysis. Right-tailed Fisher's exact test was used, and Benjamini–Hochberg (B–H) $P$ value correction performed to reduce the false discovery rate (FDR).

**Drug–target network analysis in C-AYA solid tumors**. To determine what proportion of P/LP variants detected in our dataset are harbored within potentially druggable genes, we cross-matched our gene list with the drug-target network database generated by our group[10,11]. This database contains 13,567 drug–target pairs connecting 2248 targets and 1703 US FDA-approved drugs. From 1507 patients, 511 (34%) had at least one P/LP variant on a gene that is potentially druggable. One hundred twenty-seven (8.4%) patients had KCPG P/LP variants, and an additional 384 (25.5%) individuals had P/LP variants in druggable candidate genes. About one-third of these patients (161 individuals), 72 individuals with KCPG P/LP variants, and 89 with candidate genes P/LP variants had existing FDA-approved antineoplastic and immunomodulating-related compounds. The P/LP genetic alteration of these patients was located on 73 genes, 19 KCPG, and 54 candidate genes. We had eight individuals with

two druggable genes. C-AYA patients with adrenocortical carcinoma had the highest number of patients with druggable genes (45.5%), followed by patients having sarcomas including soft tissue sarcomas (14%), Ewing sarcomas (12.6%), and osteosarcomas (11.6%). Patients with CNS tumors, retinoblastoma, Wilms tumor, and neuroblastoma were next, each with 10% of their cases carrying druggable alterations. Patients with low-grade glioma, with 4.2%, had the lowest number of individuals with druggable events (Fig. 5; Supplementary Data 19–20).

## Discussion

Despite current advancement in first-line targeted therapies for adults with solid tumors, there has not been much focus on the exploration and development of targeted treatment, specifically considering germline genomic signatures for solid tumors in

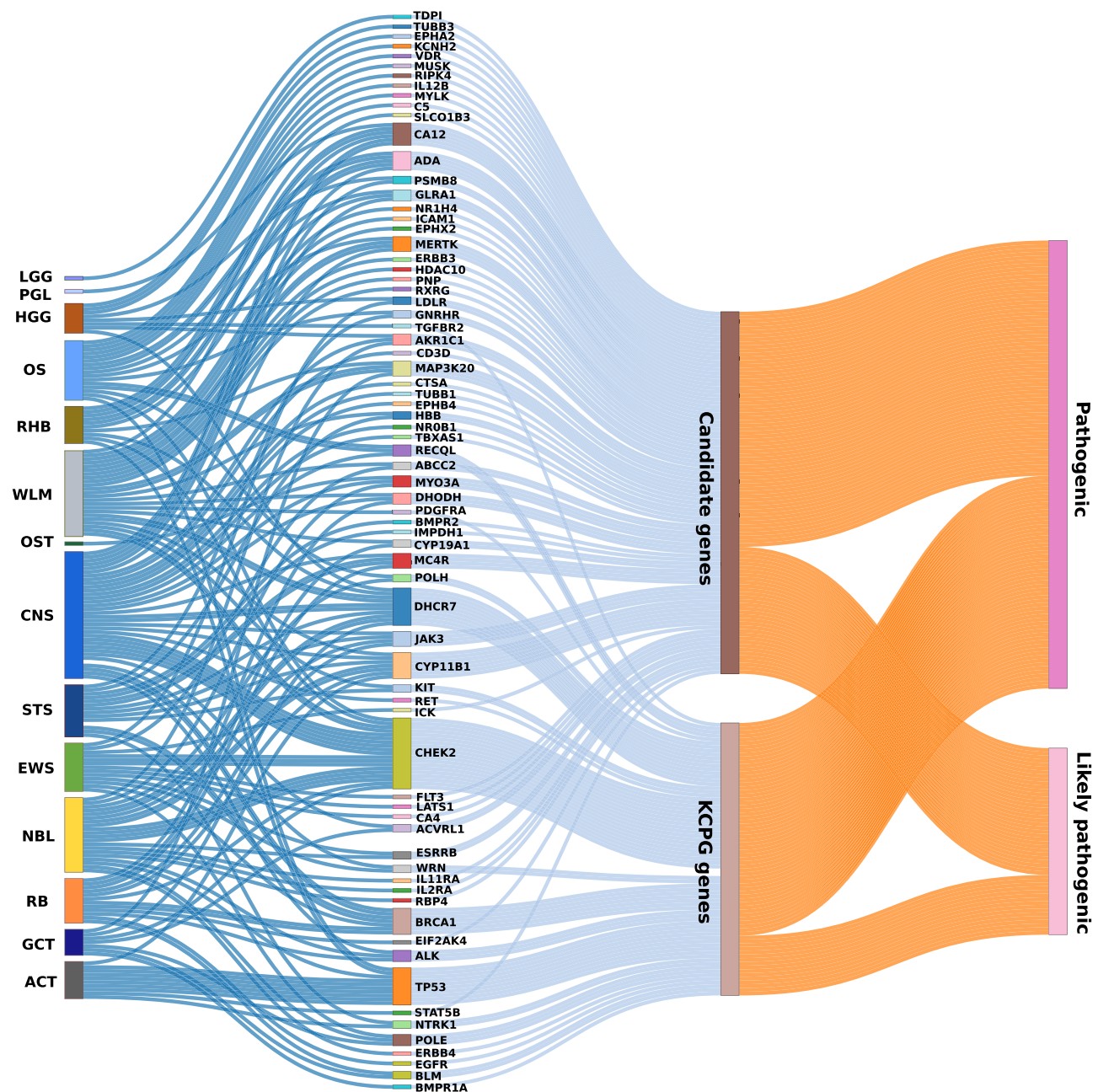

**Fig. 5 Drug–target network analysis in children, adolescents, and young adults (C-AYA) with solid tumors.** Known cancer-predisposing genes (KCPG) and candidate genes with germline pathogenic/likely pathogenic (P/LP) variants in C-AYA patients with solid tumors that have existing FDA-approved antineoplastic and immunomodulating-related compounds, in regard to their affected tumor types. ACT adrenocortical carcinoma, CNS central nervous system, EWS Ewing sarcoma, GCT germ cell tumor, HGG high-grade glioma, LGG low-grade glioma, NBL neuroblastoma, NM non-malignant tumor, OS osteosarcoma, OST other solid tumors, PGL paraganglioma, RB retinoblastoma, RHB rhabdomyosarcoma, STS soft tissue sarcoma, WLM Wilms tumor.

C-AYA patients. Grobner et al. recently published a comprehensive study in the field of C-AYA cancers, with the main focus on somatic mutational signatures. In their germline variant evaluation, they examined 162 cancer-predisposing genes in 914 C-AYA individuals with malignancy, including 798 solid tumors[1]. We expanded this evaluation to almost double the number of individuals with solid tumors (1507 C-AYA) and evaluated 204 known cancer-predisposing genes. Notably, we also expanded our assessment with an agnostic approach to evaluate new (not previously known to predispose to cancer) candidate genes, and affected pathways which added to the germline signatures of

C-AYA solid tumors. To our knowledge, this study provides the largest evaluation of germline mutations in C-AYA patients with solid tumors. Here, we performed variant-prioritization analysis on germline exome data of 1507 C-AYA patients with solid tumors, while focusing not only on the well-known germline mutations in KCPGs but also any P/LP germline alterations in genes previously unknown to be associated with cancer predisposition.

Starting with our prospectively-recruited CCF series, we showed that 10% of our cases harbored P/LP germline alterations, either a truncating mutation in a KCPG and/or a larger CNV in

cancer-related genes, consistent with the previous studies[1–3,12]. One of our CCF osteosarcoma cases presented with a germline truncating mutation in *GJB2* and duplication of *DDX10*. *GJB2*, which encodes an epithelial gap junction protein, is mostly known for being associated with syndromic hearing loss, for example, keratitis-ichthyosis-deafness (KID). It has been reported that these KID patients with germline *GJB2* mutation have increased risks of developing epithelial malignancies, for example, 19% occurrence of squamous cell carcinoma of the skin and oral mucosa compared to the normal population[13]. In total, combined with StJ cases, we detected seven *GJB2* germline P/LP variants in C-AYA patients with CNS tumors (3 patients), osteosarcoma (2), Ewing sarcoma (1), and rhabdomyosarcoma (1). *DDX10*, the second altered gene, in this case, is a known cancer-related gene, and its somatic overexpression has been recently reported to be associated with a lower survival rate in osteosarcoma patients[14]. Shi and Hao[14] showed that silencing of *DDX10* could potentially be therapeutic, and inhibit proliferation, invasion, and migration of the tumor cells, by inhibiting MAPK pathway.

Validating our findings with a larger dataset, focusing on single-nucleotide variations (SNVs) and small indels, we confirmed that 12% of C-AYA patients with solid tumors harbored at least one germline P/LP variant in the KCPGs, with an additional 61% of our cases carrying pathogenic variants in other candidate genes. As expected, about one-third of these KCPGs and candidate genes, each with four and more P/LP variants, were enriched and had statistically significant higher P/LP variant allele frequencies in our C-AYA patients with solid tumor dataset compared to the control group from non-TCGA ExAC dataset. Overrepresentation of germline P/LP variants of those genes in our dataset compared to the control dataset verified the non-incidental nature of those findings. Recently, Wang et al.[15], with the same approach, proposed that having a germline heterozygous *BRCA2* mutation predisposes to pediatric or adolescent non–Hodgkin lymphoma, by showing an overrepresentation of the *BRCA2* mutations in the target group compared to a control population without cancer (odds ratio, 3.3; 95% CI, 1.7–5.8)[15].

On the other hand, there are multiple pieces of evidence connecting a portion of these bioinformatically predicted candidate genes to cancer (Table 3). For example, somatic overexpression of *TMPRSS3*, a transmembrane serine protease, mostly known for its association with non-syndromic hearing loss, was previously reported to be associated with breast, ovarian, and pancreatic cancers[16–18]. *TMPRSS3* can mediate cancer progression, using its proteolytic activities, by helping the malignant cells to proliferate, migrate, and survive, via regulation of the ERK1/2 and PI3K/AKT pathways[19]. In another example, somatic depletion of *PRKN*, a component of a multiprotein E3 ubiquitin ligase complex, and a known gene associated with Parkinson's disease, was also reported in ovarian and lung cancers[20–23]. *PRKN* can act as a tumor suppressor gene, and its loss of function can activate the PI3K/AKT pathway via inactivation of PTEN[24]. Here, we reported multiple instances of predicted *PRKN* loss of function in patients with Wilms (6), CNS (5), neuroblastoma (4), and osteosarcoma (3) tumors (Fig. 3c, d). We also reported germline loss of function of *COL7A1* in four C-AYA patients with Wilms tumor. Interestingly, RNA expression data, comparing tumor and normal tissue from gene expression profiling interactive analysis (GEPIA)[25], confirmed the lower expression of *COL7A1* in adult kidney-related tumors as well (Supplementary Fig. 9).

There have been several epidemiologic studies associated with childhood congenital malformations with cancer risk. In a recent study from the Swedish Patient Register, e.g., Mandalenakis et al.[26] showed that C-AYA patients with any kind of CHD had increased risk of developing cancer (hazard ratio = 2.24, 95% CI, 2.01–2.48) compared to their matched controls, from Total Population Register in Sweden, who did not have CHD (2% vs. 0.9%). Here, we also showed that 67 (4.4%) of our C-AYA patients with solid tumors carried at least one germline P/LP variant in a CHD-related gene (7 KCPG and 19 candidate genes) (Supplementary Fig. 7), confirming the importance of evaluating both KCPGs and candidate genes. As examples, germline *NF1* and *PTPN11* P/LP variants were found in 24 of our C-AYA patients with solid tumors (Supplementary Data 5): both of these KCPGs predispose to CNS-related tumors, are also strongly correlated with CHDs, via up-regulation of the RAS pathway[27–30]. *NOTCH1* germline P/LP variants in two of our cases (Supplementary Data 16) is another example of a candidate gene associated with both CHD and cancer via varied mechanisms, including downregulation of the TGF-beta signaling pathway affecting epithelial-to-mesenchymal transformation (EMT)[31–33]. Together, the data to date re-emphasize the need for referring all C-AYA cancer patients for genetic consultation and further clinical evaluation.

C-AYA patients with solid tumors have a lower burden of somatic mutations while carrying a higher number of germline alterations, compared to their adult counterparts[1,2,34]. Thus, our study here reveals the germline as a therapeutic consideration. Our pathway analysis showed that not only point mutations, and deletion of *TP53* itself are important in cancer predisposition in C-AYA patients, but also that P/LP germline mutations of other, seemingly disparate, genes point to a final common disruption of the p53 pathway. Thus, the p53 signaling pathway appears to be a crucial final common pathway in cancer predisposition in C-AYA. Relatedly, we showed also that DNA damage response (DDR) and checkpoint control pathways are the top canonical pathways in this group. While it is routine to target somatic mutations in solid tumors, these observations suggest that germline mutations can also be effectively targeted in those with malignancies. The prime example is using poly (ADP-ribose) polymerase (PARP) inhibitors for the treatment of adults with advanced breast, ovarian, and prostate cancers in the context of germline mutations in DDR genes, such as *BRCA1*, *BRCA2*, *ATM*, or *PALB2*[35–37]. Although targeting mutated genes in the germline setting is challenging owing to possible toxicity to non-cancerous tissues, we speculate that appropriate drug dose thresholds could lend a high therapeutic index. Moreover, because cancer cells often have a complex network of disrupted genes and pathways (including somatic aberrations absent from normal cells), we would expect varied sensitivity to therapeutic targeting between malignant and normal cells. Our drug–target network analysis opens a new window on potentially druggable genes and possible repurposable drugs for currently considered undruggable tumor targets. Thus, further preclinical and clinical studies are warranted before translation to the routine clinical armamentarium. Acquiring and combining the data for both somatic and germline alterations, and their subsequent affected pathways, can be crucial and rudimentary in selecting the treatment strategy with the highest therapeutic index, and which may even mitigate the late effects in the C-AYA population. Towards these ends, the latest efforts by the National Cancer Institute to establish the childhood cancer data initiative (CCDI), accompanied by ongoing clinical trials such as the comprehensive omics analysis of pediatric solid tumors (NCT01109394), should collectively provide a relevant infrastructure for C-AYA solid tumors which are currently considered difficult to treat because of non-druggable targets.

Our study has several limitations, including lack of matched tumor or RNAseq data for many of our cases. Our series were not population-based cohorts; and only 5-year survivors were included in the St. Jude series. Therefore, the prioritized variants here for cancer risk may be challenged by survivor bias. Future studies (including from the St. Jude side) should fulfill these gaps.

## Methods

**Patients enrollment/sample selection**. Patients' data for this project were obtained from two sources:

1. Cleveland Clinic Foundation (CCF): 50 patients initially diagnosed under 29 years of age with a solid tumor, presenting to the Pediatric Hematology-Oncology or the Cancer Genetics Clinics, were prospectively enrolled in this study under Cleveland Clinic-approved IRB protocol 8458. Informed consent obtained from each individual participant. Final diagnosis and tumor types were confirmed by reviewing electronic medical records (EMR), including primary care physician notes, surgical notes, and pathological reports. Family history data and pedigrees were obtained by CCF genetic counselors. Any occurrence of related cancer in first and/or second-degree relatives counted as a positive family history. Patients were evaluated for the occurrence of any relapse, metastasis, second primary malignant neoplasm (SMN), or death, which collectively we classified them as patients with poor outcome.

2. St. Jude (StJ) cloud: The rest of the patients' germline/clinical data were obtained from two datasets within the St. Jude Cloud, generated by St. Jude Children Research Hospital and McDonnell Genome Institute of Washington University School of Medicine, under legal agreement 4147653:

   (a) 193 patients from the Pediatric Cancer Genome Project (PCGP)[38]
   (b) 1269 patients from St. Jude Lifetime (SJLIFE)[8]

**Sample preparation and sequencing**. Genomic DNA of CCF patients was extracted from either peripheral-blood leukocytes (ReliaPrep Large Volume HT gDNA System/Promega, Madison, WI) or buccal mucosa (DNeasy Blood & Tissue Kit/Qiagen, Germantown, MD) by standard methods at the Cleveland Clinic Genomic Medicine Biorepository (Cleveland, OH). For whole-exome sequencing on DNA samples from CCF, we used the Nextera Rapid Capture Exome library prep kit (Illumina, San Diego, CA). Samples were quantified and QC'ed using a Qubit fluorometer (Invitrogen, Carlsbad, CA) with the dsDNA broad-range assay kit and E-gel electrophoresis system (Invitrogen). Total DNA input was 50 ng, which was then sheared enzymatically. After ligating Illumina adapters and unique barcodes, libraries were validated using the Qubit dsDNA broad-range assay kit and evenly pooled using 500 ng of each tagged library. Hybridization using Illumina capture probes was completed on the final pool and amplified by PCR. Validation of the final enriched library pool was completed using the Qubit fluorometer to derive concentration (ng/μl), Bioanalyzer for library quality and average bp size, and final quantification via qPCR (KAPA Biosystems, Illumina library quantification kit). The final enriched pool was diluted, denatured, and loaded according to the standard Illumina protocols for the HiSeq 2500 system. Samples were run across two rapid run flowcells, 2 × 100 bp (paired-end) run.

**Alignments and variant calling**. Sequencing data of CCF cases were received in binary alignment map (BAM) format. We re-generated Fast-Q files, and raw reads were mapped to the human reference haploid genome sequence GRCH37/hg19 using Burrows-Wheeler Aligner (BWA v.0.6.1)[39]. Genome Analysis Toolkit (GATK 3.5)[40], Sequence Alignment/Map (SAMtools)[41], and Picard (http://broadinstitute.github.io/picard/) were used for indel-realignment, removal of PCR duplicates, and base- and quality-score recalibrations. GATK Haplotype Caller was used for variant calling of SNVs and short (<50 bp) indels.

**Variant classification**. The variant annotation and interpretation analysis for both datasets were generated through the use of Ingenuity® Variant Analysis™ (IVA) software (www.qiagenbioinformatics.com) from Ingenuity Systems (version 5.4.20190121). We kept variants with call quality at least 20.0, read depth at least 20.0, genotype quality at least 30.0, and outside top 5.0% most exonically variable 100-base windows in healthy public genomes (1000 genomes). Variants kept up to 20 bases to the intronic region if they were predicted to disrupt splicing by MaxEntScan[42]. Variants were excluded if allele frequency was greater than or equal to 1.0% in any of the following population databases: 1000 Genomes Project (phase3v5b), NHLBI ESP exomes (ESP6500SI-V2), ExAC Frequency (0.3.1), and the gnomAD Maximum Frequency (2.0.1). Variants with a Phred-scaled CADD (v1.3) score <10 (http://cadd.gs.washington.edu/info) or tolerant SIFT prediction (2016-02-23) were excluded as well unless there was an established pathogenic common variant. Subsequently, only variants that were classified as pathogenic and likely pathogenic (P/LP) by auto-classification of IVA, based on the American College of Medical Genetics and Genomics (ACMG) guidelines, were kept for further evaluation. In addition, IVA used data from the following databases for the auto-classification: Allele Frequency Community (2018-09-06), RefSeq Gene Model (2018-07-10), PolyPhen-2 (v2.2.2), PhyloP (2009-11), DbSNP (151), TargetScan (6.2), GENCODE (Release 28), CentoMD (5.0), Ingenuity Knowledge Base (Stepford 190106.000), OMIM (May 26, 2017), BSIFT (2016-02-23), TCGA (2013-09-05), ClinVar (2018-08-01), DGV (2016-05-15), COSMIC (v86), HGMD (2018.3). P/LP variants were excluded if ClinVar (https://preview.ncbi.nlm.nih.gov/clinvar/) called them benign or likely benign. All prioritized CCF variants and the

majority of indels in StJ datasets were inspected through the Integrative Genomics Viewer (IGV) to rule out artifacts[44](Supplementary Fig. 1).

**Cancer predisposition gene selection**. We compiled a list of 204 known cancer-predisposing genes (KCPG) using published literature and databases[1,2,5–7,45,46] (Cancer Gene Census Germline 2019 https://cancer.sanger.ac.uk/census#cl_search) (Supplementary Data 2). We assigned all the prioritized variants to two groups, namely, the KCPG group or the Candidate gene group based on this list. We only counted a heterozygous variant KCPG if it had a dominant inheritance pattern. None of our KCPG autosomal-recessive (AR) genes had homozygous or compound heterozygous variants.

**Variant analysis in the control population**. Case–control analysis, using 13 non-cancer patients from CCF and 340 non-cancer samples from the StJ dataset as controls, was performed to exclude pipeline alignment errors in our IVA analysis. Also, we extracted germline exome data, for all the KCPG and candidate genes found in our study, and independently performed another IVA analysis, with the same parameters, on 53,105 individuals from non-TCGA ExAC database to compare the frequency of P/LP allele variants between C-AYA patients with solid tumors and this non-cancer control population.

**CNV analysis**. We used VarSeq™ v2.1.0 (Golden Helix, Inc., Bozeman, MT, www.goldenhelix.com) to detect CNVs in our CCF dataset, using depth of coverage following the manufacturer's instructions (https://link.springer.com/protocol/10.1007%2F978-1-4939-8666-8_9). Principle component analysis (PCA) and reference sample normalization were used to normalize the data. We used two metrics to detect a CNV event: (1) $Z$-score, which is the number of standard deviations from the reference sample mean and (2) ratio, which is then normalized read depth for the sample of interest divided by the normalized mean depth over the reference samples. Both metrics are computed from normalized coverage. We used ratio ≤ 0.75 and $Z$-score ≤ −2.5 for screening of heterozygous deletion and ratio ≥ 1.25 and $Z$-score ≥ 2.5 for primary detection of the duplications. 1000 Genomes, ExAC, ClinVar, and Database of Genomic Variants (DGV) were used for data annotation. The $Z$-scores were used to compute $P$ values for each called event. eXome Hidden Markov Model (XHMM) algorithm[47] was used with default parameters to confirm our CNV findings. The mean per-target depth of coverage for detected CNVs was 149.

**Pathway analysis**. We used the OncogenicPathways function of maftools[48] to check for the enrichment of known oncogenic signaling pathways in our dataset. To calculate the fraction affected, we divided the number of genes affected in each pathway to the total number of genes within that pathway. Next, we used the Qiagen IPA[49] to generate networks for our target genes. We included only genes with four or more P/LP variants in our dataset, in KCPG and candidate genes, after optimization. Each gene ID was mapped to its corresponding object in Ingenuity's knowledge base. These genes served as seeds for generating our networks. Networks were then algorithmically generated based on the connectivity of our genes of interest with other molecules existing in the Ingenuity's knowledge base.

**Reconstruction of the drug–target network**. We collected physical drug–target interactions for FDA-approved drugs from seven commonly used data sources. Specifically, drug–target interactions were acquired from the DrugBank[50], the Therapeutic Target Database[51], the PharmGKB[52], and DrugCentral[53]. Bioactivity data of drug–target pairs were collected from three commonly used databases: ChEMBL[54], BindingDB[55], and IUPHAR/BPS Guide to Pharmacology[56]. Herein, we defined a physical drug–target interaction using the reported binding affinity/inhibitory data: inhibition constant/potency ($K_i$), dissociation constant ($K_d$), median effective concentration ($EC_{50}$), or median inhibitory concentration ($IC_{50}$), each ≤10 μM. After extracting the bioactivity data related to the drugs from the prepared bioactivity databases, only those items meeting the following four criteria were retained: (i) binding affinities, including $K_i$, $K_d$, $IC_{50}$, or $EC_{50}$, ≤10 μM; (ii) proteins represented by unique UniProt accession number; (iii) proteins marked as "reviewed" in the UniProt database[57], and (iv) proteins of human origin. In total, we collected 13,567 drug–target pairs connecting 2248 targets and 1703 US FDA-approved drugs (December 2018). We defined the therapeutic drug families based on the Anatomical Therapeutic Chemical (ATC) classification codes downloaded from DrugBank[50] and DrugCentral[53]. For example, we defined the antineoplastic and immunomodulating agents based on the first level of the ATC code as L. To select druggable genes, we cross-matched our prioritized gene list with the reconstructed drug-target network and used the Sankey diagram (R package-networkD3) for the visualization.

**Statistical analysis**. In pathway analysis, the network scores were created based on the hypergeometric distribution and were calculated with the right-tailed Fisher's exact test. B–H $P$ value correction was used to reduce the FDR. In our case–control comparison analysis, we calculated the $P$ values, ORs, and 95% CIs with a two-sided Fisher's exact test implemented in R statistical software. $P$ values were adjusted, when we compared the frequency of P/LP variants in C-AYA with solid

tumors versus non-TCGA ExAC control population, for multiple testing with Bonferroni correction considering 593 tests. FDR threshold of 0.05 considered a significant event. We used an independent dataset from St. Jude Children's Research hospital to reproduce the data from our pilot study on Cleveland Clinic series. All the analysis of this study performed multiple time to ensure the reproducibility of the findings.

**Reporting summary**. Further information on research design is available in the Nature Research Reporting Summary linked to this article.

## Data availability

The whole-exome data for C-AYA cases with solid tumors from Cleveland Clinic have been deposited in the NCBI Sequence Read Archive (SRA) database under the accession code PRJNA559601. Whole-exome data for C-AYA cases with solid tumors from St. Jude Children's Research hospital is accessible at https://www.stjude.cloud/ website. The non-TCGA data referenced during the study are available in a public repository from Broad Institute website at ftp://ftp.broadinstitute.org/pub/ExAC_release/release0.3.1/subsets/. All the other data supporting the findings of this study are available within the article and its Supplementary Information files and from the corresponding author upon reasonable request. A reporting summary for this article is available as a Supplementary Information file.

## Code availability

All data analysis, visualization, and codes related to this study are available at the following GitHub link: https://github.com/EngLabGMI/germline_caya_solidtumor_analysis.

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

## Acknowledgements

This work was supported, in part, by the VeloSano Pilot Program (Cleveland Clinic Taussig Cancer Institute). We thank St Jude Research Hospital for their generosity in making their genomic data publicly available; Dr. Peter Anderson, Kaitlin Sesock, and Ryan Noss for their help in recruiting the CCF patients; Phyllis Harbor for her help with biorepositing and processing the samples; Dr. Eduardo Pérez for helpful statistical discussions; and Dr. Ying Ni for helpful bioinformatic advice. L.Y. is an Ambrose Monell Foundation Cancer Genomic Medicine Fellow at the Cleveland Clinic Genomic Medicine Institute. C.E. is the Sondra J. and Stephen R. Hardis Chair of Cancer Genomic Medicine at the Cleveland Clinic and an American Cancer Society Clinical Research Professor.

## Author contributions

Conceptualization: S.A., L.Y., and C.E.; methodology: S.A., L.Y., R.P., and C.E.; mutation analysis & investigation: S.A.; drug-target network analysis: F.C.; data curation and bioinformatic analysis & visualization: R.P. and S.A.; validation, supervision & funding acquisition: C.E.; writing—original draft, review & editing: S.A., L.Y., F.C., and C.E. All authors critically reviewed and approved the final manuscript.

## Competing interests

The authors declare no competing interests.
