## [Peer Review File · Nature Communications]

Editorial Note: This manuscript has been previously reviewed at another journal that is not operating a transparent peer review scheme. This document only contains reviewer comments and rebuttal letters for versions considered at *Nature Communications*

Reviewers' comments:

Reviewer #1 (Remarks to the Author):

The authors have made a number of important changes in response to this reviewer's comments and in many ways have addressed some key concerns (and particular clarifications). However, there are still a number of points that are of some concern:

1. the title of the manuscript focuses on the convergence of variants on the p53 network. However, the bulk of the findings and focus particularly in the discussion relates to other observations and interesting gene associations. So I am not sure the title truly reflects the most salient points of the manuscript.
2. Throughout the manuscript (and in particular in the sentences and phrases added in the new text) there are numerous grammatical and syntax errors that should be attended to. For some of these, the meaning of the sentence or phrase becomes lost as a result.
3. As per point 2, the new wording in response to Reviewer 1, comments 6 and 7 are difficult to figure out
- 4 the response to point 8 is a bit challenging. I'm not sure I agree that germline variants are necessarily a 'better' biomarker and therapeutic target; they are just different. In fact, as the authors point out later, targeting in the context of heterozygous variants in p53 pathway or DNA damage repair genes raises other significant challenges such as activating dominant-negative effects with anticipated significant toxicities.
5. The response to point 11 is ok - except that it should be noted in the text itself - that is, demonstrate how this builds on the Grobner work. As well, the paucity of targeted treatments in CAYA cancers has as much to do with complex regulatory issues around drug development and small populations to study on trials, as it does to the actual targets.
6. I think that without the clinical data (which understandably is difficult to obtain) on the GJB2 and CHD variants stories, it is difficult to really speculate on what these findings really mean in the context of this manuscript. At best, an association is being made - but whether it is actually relevant is beyond the scope of the findings as reported.
7. the statement in response to comment 14 is still actually in the text. It is still line 347-348. This should be removed.
8. the authors' response to comment 15 is not quite complete, in that we really actually do not know what would happen with reactivation or increased WT p53 expression in the context of heterozygosity - may be dependent on the actual variant, etc. Nonetheless, I agree with taking out that line as the authors have done.

Reviewer #4 (Remarks to the Author): Replacement for Reviewer #2

Diverse germline variants in C-AYA cancers

This is an interesting paper with potentially important results, delineating germline variants that converge on the p53 pathway and RAS pathways. A small amount of new data is combined with large existing germline data (for reanalysis), attempting to delineate the scope of germline variants that may impact disease risk and outcome. The analysis is thorough but there are some concerns in design and interpretation.

Major comments

There are two possible caveats to be concerned about for the potential significance of the data.

First, the sequencing technology was different for the test samples and the controls (derived from existing databases). This has the potential effect of revealing false positives since there are systematic sequencing errors with different kits, protocols, exome coverage and sequence depth. Some discussion on this issue is needed. Second, EXAC was used as a comparison, which is a small comparison group when there are other larger control groups – for instance the gnomAD – in addition there is little consideration of ethnic heterogeneity among the populations studied and therefore the mutation frequency comparisons. This ethnic heterogeneity is better controlled in the larger population-based databases such as gnomAD. It is quite difficult to know whether children with cancer in these cohorts may harbor rare alleles simply by being members of racial/ethnic groups poorly represented in the controls particularly with LP variants which are very common in non-diseased populations of all ethnic groups. While ethnic groups are not known in the database of most subjects (from the excel file submitted), this will be discernable from the genetic data.

The majority of patients are from the STJudes cohort, which is not a population-based cohort of samples; and furthermore, only 5 year survivors were sequenced from that cohort. Therefore the variant estimates here for cancer risk are going to be deeply challenged by survivor bias from that cohort study. This needs to be acknowledged. Similarly, the PCGP involves a good deal of patient and tumor type selection for sequencing. In both of these studies, far more samples and data was available than was analyzed here in this study. Is that due to availability of germline data issues, data quality filters, or other reasons? This is not stated in Methods.

A fuller comparison on inherited versus germline mutation for the variants tested would be helpful. There is limited discussion on family history.

The abstract is misleading, suggesting that 1500+ patients were sequenced, when this is largely a re-analysis of St Judes sequencing data and only 50 new patients.

Was there any clinical follow up on the 50 new patients? This seems to be limited to Fig 1E – were any significant results shown? Was this cohort similar or distinct from the STJudes Life and PCGP in spectrum of diagnoses and outcomes?

Discussion: The focus on “druggable” may be inappropriate. While these genes may be involved in the etiology of the disease, much of that effect will involve the higher propensity to obtain new somatic variants within the cell clone that becomes a tumor. These latter somatic variants (not assessed here) will more likely be relevant drug targets, not the mutator germline variants that helped facilitate those somatic mutations. Instead, many of these germline variants may support genetic counseling and further sequencing for the family members of the proband with cancer. As the authors note, germline variants will affect all tissues in an organism and any therapies will hence affect all tissues – the hope that this will be solved by titratable therapeutic indices is possible in some cases but will be very difficult when there are so few patients with such mutations for clinical trial considerations.

NCOMMS-19-34930-T: Responses to the Reviewers' Comments

We thank the editors and peer reviewers for their insightful and encouraging feedback that has helped lift our revised manuscript in both content and clarity, and thanks both editors for the opportunity to transfer to NATCOMM.

In detailed response to the **REVIEWER COMMENTS:**

REVIEWER #1:

We thank this reviewer for taking time out to meticulously review our responses and for acknowledging that “The authors have made a number of important changes in response to this reviewer’s comments and in many ways have addressed some key concerns (and particular clarifications)”. In detailed response to his/her comments:

SPECIFIC POINTS

1. The title of the manuscript focuses on the convergence of variants on the p53 network. However, the bulk of the findings and focus particularly in the discussion relates to other observations and interesting gene associations. So, I am not sure the title truly reflects the most salient points of the manuscript.

RESPONSE:

We thank the reviewer for bringing up this point. Our work suggests P/LP germline variants in a broad range of genes (both known cancer predisposition and candidate genes) in individuals with C-AYA solid tumors. Despite the apparent breadth of diversity, our pathway analysis pointed to a final common disruption of the p53 pathway. This is extremely important and shows that not only point mutations and deletion of *TP53* itself are important in cancer predisposition in C-AYA patients, but also that P/LP germline mutations of other genes in this pathway can also predispose individuals to C-AYA solid tumors. We tried to combine these two observations in our title, as this is the take home message of this manuscript. Thus, we respectfully request that we keep our title as is but are open to advisement from the reviewers and editor, if otherwise.

2. Throughout the manuscript (and in particular in the sentences and phrases added in the new text) there are numerous grammatical and syntax errors that should be attended to. For some of these, the meaning of the sentence or phrase becomes lost as a result.

RESPONSE:

We thank the reviewer for bringing this to our attention and have tried our best to address this issue throughout the manuscript.

3. As per point 2, the new wording in response to Reviewer 1, comments 6 and 7 are difficult to figure out

RESPONSE: We apologize for our lack of clarity and thank the reviewer for bringing up this point. We have rewritten the two paragraphs as follows:

page 3, line 62 (all line numbers reference the single PDF version): “In general, pathogenic loss-of-function variants in tumor suppressor genes or gain-of-function variants in oncogenes can disrupt normal cellular processes, and predispose to cancer development⁶; moreover, multiple genes can have both types of variants with different functional and phenotypic effects. It is well established that there is an overlap between germline cancer-predisposing genes and somatic tumor-driver genes: there are many examples showing identical genes having roles in somatic oncogenesis and susceptibility to cancer, respectively⁷.”

page 3, line 69: “In pediatric and AYA clinics, family history is essentially the primary means used to recognize patients with possible heritable cancer⁸”

4. The response to point 8 is a bit challenging. I'm not sure I agree that germline variants are necessarily a 'better' biomarker and therapeutic target; they are just different. In fact, as the authors point out later, targeting in the context of heterozygous variants in p53 pathway or DNA damage repair genes raises other significant challenges such as activating dominant-negative effects with anticipated significant toxicities.

RESPONSE:

We appreciate the reviewer's comment and have edited our sentences removing any comparison of betterment. We edited this part in the manuscript (**page 4, line 74**) as follows: “In addition to implications for heritability and second primary neoplasms, germline variants can also provide novel therapeutic targets. The clonal nature of germline (in every single cell of the body) variants compared to the heterogeneous somatic pattern of tumors make them potentially a suitable biomarker and therapeutic target, both of which are lacking for C-AYA malignancies, compared to adult malignancies. Here, we address this gap by investigating germline genomic signatures of 1,507 patients with solid tumors diagnosed under 29 years of age.”

We respectfully believe that this reviewer may have misunderstood our dominant negative discussion point. We essentially are saying the same thing. What we are saying is that instead of targeting p53 itself, we now have multiple other genes and pathways that lead to p53 and hence, have other potentially targetable proteins and pathways.

5. The response to point 11 is ok - except that it should be noted in the text itself - that is, demonstrate how this build on the Grobner work. As well, the paucity of targeted treatments in CAYA cancers has as much to do with complex regulatory issues around drug development and small populations to study on trials, as it does to the actual targets.

RESPONSE:

We thank the reviewer for bringing up this point. We updated the text as follows:

page 11, line 278: “Grobner et al., recently published a comprehensive study in the field of C-AYA cancers, with the main focus on somatic mutational signatures. In their germline variant evaluation, they examined 162 cancer-predisposing genes in 914 C-AYA individuals with malignancy including 798 solid tumors⁴. We expanded this evaluation to almost double

the number of individuals with solid tumors (1507 C-AYA) and evaluated 204 known cancer-predisposing genes. Notably, we also expanded our assessment with an agnostic approach to evaluate new (not previously known to predispose to cancer) “candidate genes” and affected pathways which added to the germline signatures of C-AYA solid tumors.”

6. I think that without the clinical data (which understandably is difficult to obtain) on the GJB2 and CHD variants stories, it is difficult to really speculate on what these findings really mean in the context of this manuscript. At best, an association is being made - but whether it is actually relevant is beyond the scope of the findings as reported.

RESPONSE:

We thank the reviewer for bringing up this point. In fact, we agree with your point and our goal was just showing the associations and bringing attention to these genes/conditions and for seeding future studies to evaluate the causalities. However, in the meantime, referring these C-AYA cancer patients for genetic consultation and further clinical evaluation could be beneficial to both patients and their family members and we should not overlook that.

7. The statement in response to comment 14 is still actually in the text. It is still line 347-348. This should be removed.

RESPONSE:

We apologize for this omission. We removed that sentence from discussion (**Page 14, line 352**).

8. The authors' response to comment 15 is not quite complete, in that we really actually do not know what would happen with reactivation or increased WT p53 expression in the context of heterozygosity - may be dependent on the actual variant, etc. Nonetheless, I agree with taking out that line as the authors have done.

RESPONSE:

We thank the reviewer for bringing up this point and apologize for not being clear. Targeting p53 is still challenging and as mentioned, reactivation of or increasing WT p53 expression in the presence of a heterozygous missense mutation can result different outcomes depending on the actual variant. We are glad that removing that sentences reduced the confusion (**Page15, line 374**).

REVIEWER #4 (REPLACEMENT FOR REVIEWER #2):

We thank this reviewer for taking time out to meticulously review our responses and for acknowledging that “This is an interesting paper with potentially important results”, and “The analysis is thorough“. In detailed response to his/her comments:

SPECIFIC POINTS

1. There are two possible caveats to be concerned about for the potential significance of the data. First, the sequencing technology was different for the test samples and the controls (derived from existing databases). This has the potential effect of revealing false positives since there are systematic sequencing errors with different kits, protocols, exome coverage and sequence depth. Some discussion on this issue is needed. Second, EXAC was used as a comparison, which is a small comparison group when there are other larger control groups – for instance the gnomAD – in addition there is little consideration of ethnic heterogeneity among the populations studied and therefore the mutation frequency comparisons. This ethnic heterogeneity is better controlled in the larger population-based databases such as gnomAD. It is quite difficult to know whether children with cancer in these cohorts may harbor rare alleles simply by being members of racial/ethnic groups poorly represented in the controls particularly with LP variants which are very common in non-diseased populations of all ethnic groups. While ethnic groups are not known in the database of most subjects (from the excel file submitted), this will be discernable from the genetic data.

RESPONSE: We thank the reviewer for bringing up these important points. As we previously mentioned in our methods (**Page 19, Line 467**), in order to exclude all the possible pipeline alignments, and systematic sequencing errors, we primarily performed two separate case-control Ingenuity Variant Analysis (IVA), using 13 non-cancer patients from CCF and 340 non-cancer samples from the StJ dataset as controls, and filtered out any common variants between case and control groups and focused only on the variants which were specific to our cancer patients. Since in each group, both cancer patients and their non-cancer controls went through identical sequencing and pipeline methods, we are confident that the majority of the mentioned potential false-positive variants should be already excluded from the final analysis.

Regarding the second point, as we previously mentioned in our methods (**Page 18, Line 441**), “Variants were excluded if the allele frequency was greater than or equal to 1.0% in any of the following population databases: 1000 genomes project (phase3v5b), NHLBI ESP exomes (ESP6500SI-V2), ExAC Frequency (0.3.1), and the gnomAD Maximum Frequency (2.0.1)”. So, we looked at both ExAC and gnomAD databases and excluded any common variants in the population. As mentioned, our ethnic data was limited, and we could not further stratify our data based on the ethnicity of our patients.

In a further analysis, we extracted germline exome data, for all the KCPG and candidate genes found in our study, and independently performed another IVA analysis, with the same parameters, on 53,105 individuals from non-TCGA ExAC database to compare the frequency of P/LP allele variants between C-AYA patients with solid tumors and this non-cancer control population. In other words, we compared the enrichment of P/LP variants in those genes between our C-AYA patients with solid tumors and this non-cancer control population. By the time we performed this analysis, ExAC was the only database that provided the full possibility of downloading the non-TCGA subset of its data for additional analysis.

2. The majority of patients are from the STJudes cohort, which is not a population-based cohort of samples; and furthermore, only 5-year survivors were sequenced from that cohort. Therefore,

the variant estimates here for cancer risk are going to be deeply challenged by survivor bias from that cohort study. This needs to be acknowledged. Similarly, the PCGP involves a good deal of patient and tumor type selection for sequencing. In both of these studies, far more samples and data was available than was analyzed here in this study. Is that due to availability of germline data issues, data quality filters, or other reasons? This is not stated in Methods.

RESPONSE: We thank the reviewer for bringing up these important points. We updated the text and added this limitation for the clarification:

Page 16, Line 383: “Our study has several limitations, including lack of matched tumor or RNAseq data for many of our cases. Our series were not population-based cohorts; and only 5-year survivors were included in the St Jude series. Therefore, the prioritized variants here for cancer risk may be challenged by survivor bias. Future studies (including from the St. Jude side) should fulfill these gaps. “

For both studies, we downloaded the data for patients who had a diagnosis of a solid tumor and had germline data in the format of gvcf available from a whole-exome sequencing analysis. All the detailed coding data are provided in our GitHub link.

3. A fuller comparison on inherited versus germline mutation for the variants tested would be helpful. There is limited discussion on family history.

RESPONSE: We appreciate the reviewer’s comment regarding the importance of this matter. Since we did not have a Trio-study on any of our patients, we could not discuss further the inheritance of these germline variants. This is due to IRB and regulatory limitations for the St Jude dataset. But we agree with the reviewer that having those data could be very useful to our discussion. For our own institutional our CCF series, we of course had family history as noted: “Family history data and pedigrees were obtained by CCF genetic counselors. Any occurrence of related cancer in 1st and/or 2nd-degree relatives counted as a positive family history” as it mentioned in our methods (**Page 16, Line 396**). Our data show the existence of a positive family history of cancer in about 40% of the KCPG mutation-negative patients and is noted in the manuscript. This encouraged us to look for new candidate predisposing genes in this group of patients.

4. The abstract is misleading, suggesting that 1500+ patients were sequenced, when this is largely a re-analysis of St Judes sequencing data and only 50 new patients.

RESPONSE: We appreciate the reviewer’s comment regarding this matter. In fact, in our abstract, we specifically mentioned “variant-prioritization analysis on germline DNA of 1,507 C-AYA patients with solid tumors” and not the sequencing. But if the reviewer thinks this is still misleading, we would be happy to re-write this part.

5. Was there any clinical follow up on the 50 new patients? This seems to be limited to Fig 1E – were any significant results shown? Was this cohort similar or distinct from the STJudes Life and PCPG in spectrum of diagnoses and outcomes?

RESPONSE: We thank the reviewer for bringing up this point. Yes, we retrospectively reviewed the electronic medical records (EMR) of the 50 CCF patients. We evaluated the occurrence of any relapse, metastasis, second primary malignant neoplasm (SMN), or death, which collectively we classified them as patients with poor outcome and summarized the results in Figure 1E. Due to a small sample size, we could not statistically show a difference (as mentioned in the text) and just showed a trend toward a “poor outcome” in C-AYA patients with germline alterations compared to the group without germline alterations.

We had no outcome data from St. Jude cohort (again, due to regulatory and access limitations), so we cannot compare the outcome between the two series.

6. Discussion: The focus on “druggable” may be inappropriate. While these genes may be involved in the etiology of the disease, much of that effect will involve the higher propensity to obtain new somatic variants within the cell clone that becomes a tumor. These latter somatic variants (not assessed here) will more likely be relevant drug targets, not the mutator germline variants that helped facilitate those somatic mutations. Instead, many of these germline variants may support genetic counseling and further sequencing for the family members of the proband with cancer. As the authors note, germline variants will affect all tissues in an organism and any therapies will hence affect all tissues – the hope that this will be solved by titratable therapeutic indices is possible in some cases but will be very difficult when there are so few patients with such mutations for clinical trial considerations.

RESPONSE: We thank the reviewer for bringing up these important points. We agree to be conservative because targeting germline variants is very challenging and relatively new compared to targeting somatic variants. There are examples of targeting germline variants: in addition to those cited in our Discussion (adult onset cancer examples), examples of targeting the germline include the highly successful use of mTORi in those with germline *TSC1/2* mutations, and currently, we have an everolimus trial of those with germline *PTEN* mutations. Both these situations, including children as participants, have shown that targeting germline variation is well tolerated. We wanted to bring out in discussion that C-AYA tumors have much less somatic variants compared to adult tumors, and that is one of the other limitations toward targeted therapies in this population. Thus, we wanted to discuss that perhaps considering germline P/LP variation to target may amplify the number of targets (we have added to **Pages 14-15, Lines 356-361**) “Thus, our study here reveals the germline as a therapeutic consideration.” It is important to acknowledge that due to harboring germline alterations, early age of tumor onset, and possibly more prolonged survival, the C-AYA population are more prone to secondary malignant neoplasms compared to adults with cancers. So, targeting germline variants, as a potential preventive targeted therapy, although challenging, can be beneficial for this population and worth pursuing. PARP-inhibitors (as mentioned in the text) are good examples of these promising new targeted therapies, in our context that the DDR/p53 pathway appears prominent in the germline of the C-AYA population.

In regards to the second point, we also believe in the power of genetic counseling, surveillance studies, and screening for the patients with germline alterations and their family members and discussed it thoroughly in the manuscript.

Note: Line numbers are related to the single PDF file.

REVIEWERS' COMMENTS:

Reviewer #4 (Remarks to the Author):

Review

The authors thoughtfully responded to comments and altered the manuscript in appropriate fashion. There are only a few comments to add below.

"KCPG" in abstract is an acronym without explanation, also "P/LP" Such acronyms need to be spelled out probably in the abstract.

I agree with the other reviewer who suggested a title that was more inclusive of the wide range of gene alterations discussed would be more helpful, and attract a wider readership, including the congenital heart defect community. If the title was not changed, at least the abstract has a statement about the interesting "candidate" genes which emerged from this analysis – can the word "congenital" be added somewhere so pubmed searches would capture this paper.

The first sentence is appropriate for thinking about adult cancer, but not so much for childhood cancer where epigenetic modifications and developmental miscues seem more critical than an accumulated "wear and tear." This sentence should be changed.

Second sentence: The word "environment" does not appear in reference #1 and the second sentence of the introduction seems specious. There is no evidence that environment is unimportant for childhood cancer, in fact children are far more sensitive to developmental aberrations from environmental influences than adults, who have terminated organ development. This is an absurd statement to start this manuscript, and the cited manuscript (ref 1) explains several environments that have a strong impact on cancer incidence such as hormones but that reference is generally focused on therapy and outcomes.

NCOMMS-19-34930A: Responses to the Reviewers' Comments

We thank the editors and peer reviewer #4 for their insightful and encouraging feedback that has helped lift our revised manuscript in both content and clarity. We paid particular attention to the editorial requests which we believe we have addressed as detailed below as well as our cover letter.

In detailed response to the **REVIEWER COMMENTS:**

REVIEWER #4:

We thank this reviewer for taking time out to meticulously review our responses and for acknowledging that “The authors thoughtfully responded to comments and altered the manuscript in appropriate fashion”. In detailed response to his/her comments:

SPECIFIC POINTS

1. “KCPG” in abstract is an acronym without explanation, also “P/LP” Such acronyms need to be spelled out probably in the abstract.

RESPONSE: We have spelled out both acronyms and defined them in abstract and text and edited to keep the abstract word count to 150 as well.

page 2, line 29: “Performing variant-prioritization analysis on germline DNA of 1,507 C-AYA patients with solid tumors, we show 12% of these patients carrying germline pathogenic and/or likely pathogenic variants (P/LP) in known cancer-predisposing genes (KCPG). An additional 61% have germline pathogenic variants in non-KCPG genes, including PRKN, SMARCAL1, SMAD7, which we refer to as “candidate” genes. Despite germline variants in a broad gene spectrum, pathway analysis leads to top networks centering around p53. Our drug-target analysis shows 1/3 of patients with germline P/LP variants have at least one druggable alteration, while more than half of them are from our “candidate” gene group, which would otherwise go unidentified in routine clinical care.”

2. I agree with the other reviewer who suggested a title that was more inclusive of the wide range of gene alterations discussed would be more helpful, and attract a wider readership, including the congenital heart defect community. If the title was not changed, at least the abstract has a statement about the interesting “candidate” genes which emerged from this analysis – can the word “congenital” be added somewhere so pubmed searches would capture this paper.

RESPONSE: We appreciate the reviewer’s comment and have edited our title to a be more inclusive as follows. However, we also needed to be within the 15-word limit in a title

Title: Comprehensive germline genomic profiles of children, adolescents, and young adults with solid tumors

We will also ensure including the word “Congenital” to our keywords for submission. Due to the 150-word limitation of Abstract, we had to restrict ourselves to more prioritized results of the study.

3. The first sentence is appropriate for thinking about adult cancer, but not so much for childhood cancer where epigenetic modifications and developmental miscues seem more critical than an accumulated “wear and tear.” This sentence should be changed. Second sentence: The word “environment” does not appear in reference #1 and the second sentence of the introduction seems specious. There is no evidence that environment is unimportant for childhood cancer, in fact children are far more sensitive to developmental aberrations from environmental influences than adults, who have terminated organ development. This is an absurd statement to start this manuscript, and the cited manuscript (ref 1) explains several environments that have a strong impact on cancer incidence such as hormones but that reference is generally focused on therapy and outcomes.

RESPONSE: We thank the reviewer for bringing up this point and apologize for not being clear. We completely removed the first few confusing sentences and replaced them by following sentences:

page 3, line 50: “Solid tumors account for half of the malignancies in children, adolescents, and young adults (C-AYA), with lower burden of somatic variants, and are assumed to have higher frequencies of germline alterations, compared to adult solid tumors 1,2. Although there has been substantial advancement in understanding somatic variants in cancers, our knowledge regarding the spectrum, frequency, and implications of germline variants in C-AYA with solid tumors is limited.”